# Differential Effects of Gold Nanoparticles and Ionizing Radiation on Cell Motility between Primary Human Colonic and Melanocytic Cells and Their Cancerous Counterparts

**DOI:** 10.3390/ijms22031418

**Published:** 2021-01-31

**Authors:** Elham Shahhoseini, Masao Nakayama, Terrence J. Piva, Moshi Geso

**Affiliations:** 1Discipline of Medical Radiation, School of Health and Biomedical Sciences, RMIT University, Bundoora, VIC 3083, Australia; s3523192@student.rmit.edu.au (E.S.); masao.nakayama@rmit.edu.au (M.N.); 2Discipline of Human Bioscience, School of Health and Biomedical Sciences, RMIT University, Bundoora, VIC 3083, Australia; terry.piva@rmit.edu.au

**Keywords:** gold nanoparticles, ionizing radiation, cell migration, human colon epithelial cells, human colorectal adenocarcinoma cells, human epidermal melanocytes, human primary melanoma cells, radiotherapy

## Abstract

This study examined the effects of gold nanoparticles (AuNPs) and/or ionizing radiation (IR) on the viability and motility of human primary colon epithelial (CCD841) and colorectal adenocarcinoma (SW48) cells as well as human primary epidermal melanocytes (HEM) and melanoma (MM418-C1) cells. AuNPs up to 4 mM had no effect on the viability of these cell lines. The viability of the cancer cells was ~60% following exposure to 5 Gy. Exposure to 5 Gy X-rays or 1 mM AuNPs showed the migration of the cancer cells ~85% that of untreated controls, while co-treatment with AuNPs and IR decreased migration to ~60%. In the non-cancerous cell lines gap closure was enhanced by ~15% following 1 mM AuNPs or 5 Gy treatment, while for co-treatment it was ~22% greater than that for the untreated controls. AuNPs had no effect on cell re-adhesion, while IR enhanced only the re-adhesion of the cancer cell lines but not their non-cancerous counterparts. The addition of AuNPs did not enhance cell adherence. This different reaction to AuNPs and IR in the cancer and normal cells can be attributed to radiation-induced adhesiveness and metabolic differences between tumour cells and their non-cancerous counterparts.

## 1. Introduction

Ionizing radiation (IR) has been widely used clinically in the treatment of a broad range of cancers over the past few decades [1]. Cancer treatment techniques are focused on minimising the viability and migration (metastasis) of cancer cells. In radiation therapy, the effects of IR on the cells surrounding the tumour are of concern. Even though there is increased accuracy of targeting the cancerous tissues, those cells surrounding the tumour or in the path of the IR beam can become exposed to therapeutic radiation [2,3]. Therefore, it is vital to understand the mechanisms underlying the effects of IR on the viability and motility of both cancer and non-cancerous cells to improve these treatment techniques.

While IR has proven to be an effective tool in decreasing the viability of cells in a tumour mass [4], its effects on cell migration remain controversial so that both promoting and inhibiting effects have been attributed to it [5].

Different methods have been examined to improve radiation therapy by exploiting their radiation-induced cancer cell-killing while minimising the migration of these cells. One of these recent methods is the enclosure of metallic nanoparticles such as gold nanoparticles (AuNPs) in the cancer cell. The findings of such studies have shown that AuNPs have inhibitory effects on cancer cell migration without impacting on the viability of these cells which makes it a promising candidate for further application in gold nanoparticle aided radiotherapy (GNRT) [6,7,8,9]. Additionally, metallic nanoparticles have been used to enhance other cancer treatments methods such as magnetic hyperthermia and phototherapy [10,11,12].

Despite extensive studies on the effects of IR and/or AuNPs on the viability and motility of cancer cells, very little is published about the percentage viability and relative migration of the corresponding non-cancerous cells when a solid tumour mass is exposed to a therapeutic IR dose in the presence or absence of AuNPs.

Based on the metabolic differences between non-cancerous and cancer cells [13], in this in vitro study, we hypothesized that the same therapeutic dose of X-rays and the same concentration of AuNPs in the cells should result in different cell migration rates between tumour cells and their non-cancerous counterparts.

In order to be closer to the real radiation therapy conditions in both colon and melanoma cancers in which the normal tissues surrounding the tumour or in the path of the IR beam are exposed to the same dose of radiation, all the experiments were conducted using human SW48 colorectal adenocarcinoma and MM418-C1 melanoma cell lines and the results were compared to their corresponding non-cancerous counterparts, i.e., human CCD841 colon epithelial cells and epidermal melanocytes (HEM). In addition, due to dose enhancement and migration inhibiting effects of AuNPs combined with IR on cancer cells and based on the findings in our previous work [14], the effects of 15 nm AuNPs combined with 5 Gy therapeutic X-ray on the viability and motility of these cancer cells and their non-cancerous counterparts were investigated.

## 2. Results

### 2.1. Cellular Uptake of Gold Nanoparticles (AuNPs)

Inductively coupled plasma mass spectrometry (ICP MS) was used to determine the picograms (pg) of Au atoms that were accumulated per cell for human colorectal adenocarcinoma (SW48), colon epithelial (CCD841) cells, melanoma (MM418-C1), and human epidermal melanocytes (HEM). The cells were incubated with different concentrations (control (0), 0.25, 1, 2, and 4 mM) of AuNPs for 24 h prior to ICP MS measurement. As expected, in cells exposed to the control media (0 mM AuNPs) no gold was detected (Figure 1). The level of gold that accumulated in all the cells was proportional to the external media concentration. Both non-cancerous cells (CCD841 and HEM) accumulated similar levels of Au and this was significantly higher than that of their respective cancerous counterparts (SW48 and MM418-C1). The cellular uptake of 15 nm AuNPs by both cancer cell lines, i.e., SW48 and MM418-C1 approached saturation at 1 mM. However, no saturation level was observed in normal cell lines up to 4 mM for AuNPs. Based on these results, the concentration of 1 mM of AuNPs was chosen for subsequent experiments.

### 2.2. Effect of AuNPs on Cell Viability

The effect of 48 h exposure to different concentrations (0–4 mM) of 15 nm AuNPs on the viability of SW48, CCD841, MM418-C1 and HEM cells was measured using the (3-(4,5-dimethylthiazol-2-yl)-5-(3-carboxymethoxyphenyl)-2-(4-sulfophenyl)-2H-tetrazolium, inner salt) MTS assay and the results shown in Figure 2. Even at 4 mM, the AuNPs had no effect on the viability of the cell lines that were tested. At some concentrations tested it appeared that AuNPs displayed a hormetic effect; however, this was not significant.

### 2.3. Effect of Ionizing Radiation (IR) on Cell Viability

The effect of ionizing radiation (IR) on the viability of SW48, CCD841, MM418-C1 and HEM cells was determined by exposing these cells to different doses (0–6 Gy) of 6 MV X-rays. These dose ranges were chosen based on the standard dose fractionation regimen that is commonly used in radiation therapy [15]. Cell viability was measured after 48 h post-irradiation using the MTS assay. As seen in Figure 3, the viability of both cancer and their non-cancerous counterparts appeared to be proportional to the radiation dose. Both cancer cell lines displayed a significantly higher sensitivity to the radiation particularly to doses greater than 3 Gy. At 6 Gy, the viability of both non-cancerous cell lines (CCD841 prostate epithelial cells and HEM epidermal melanocytes) were ~15% higher when compared to their cancerous counterparts (SW48 colorectal adenocarcinoma and MM418-C1 melanoma cells).

### 2.4. Effect of AuNPs and Ionizing Radiation (IR) on Cell Viability

The effect of AuNPs and IR on SW48, CCD841, MM418-C1 and HEM cell viability was examined by treating the cells with 1 mM AuNPs 24 h prior to being exposed to different doses (0–6 Gy) of 6 MV X-rays. Cell viability was measured 48 h post-irradiation using the MTS assay (Figure 4). As seen, AuNPs did not increase the cytotoxic effects to that of IR alone on either cell type as seen earlier (Figure 3).

### 2.5. Effect of AuNPs on Cell Migration

The effect of AuNPs on cell migration was determined by measuring the closure of a gap created by a 200 µL pipette tip on cell monolayers grown in 6-well plates. The cells were incubated with 1 mM of AuNPs (15 nm in size) 24 h prior to the formation of the scratch. The closure of this gap (scratch area) was observed over 24 h using the CytoSmart^®^ Live Image system. At the end of 24 h the effect of the AuNPs on the size of the gap for each cell line was compared to that of its corresponding untreated control which was given the value of 100%. As seen in Figure 5, treatment with 1 mM AuNPs for 24 h retarded the migration of both cancer cells, i.e., SW48 and MM418-C1 by ~20%, while the migration of their corresponding non-cancerous cell lines CCD841 and HEM were enhanced by ~13%.

### 2.6. Effect of IR on Cell Migration

The effect of IR on cell migration was determined by examining the closure of a gap created by a 200 µL pipette tip on cell monolayers grown in 6-well plates. Cells were exposed to 5 Gy of 6 MV X-ray 24 h prior to the formation of the scratch. The closure of this gap (scratch area) was observed over 24 h using the CytoSmart^®^ Live Image system. At the end of 24 h the effect of IR on the size of the gap for each cell line was compared to that of its corresponding untreated control which was given the value of 100%. As seen in Figure 6, prior exposure to 5 Gy X-rays for 24 h retarded the migration of both cancer cells, i.e., SW48 and MM418-C1 by ~15% while that of both corresponding non-cancerous cell lines CCD841 and HEM was enhanced by ~16%.

### 2.7. Effect of AuNPs and IR on Cell Migration

The effect of 1 mM AuNPs and 5 Gy irradiation on the migration of these cells was examined to observe if a synergistic effect occurs. A 200 µL sterile pipette tip was used to make a gap in confluent monolayers of SW48, CCD841, MM418-C1 and HEM cells grown in 6-well plates. The cells were treated with 1 mM AuNPs 24 h prior to being irradiated with 5 Gy of 6 MV X-ray prior to being scratched. The closure of this gap (scratch area) was observed over 24 h using the CytoSmart^®^ Live Image system. At the end of 24 h the effect of IR + AuNPs on the size of the gap for each cell line was compared to that of its corresponding untreated control which was given the value of 100%.

As shown in Figure 7, combination of AuNPs and IR had an additive suppression effects on cell migration rate in both cancer cell lines, i.e., SW48 and MM418-C1 where a ~35% reduction in gap closure was observed. Surprisingly, in the non-cancerous cells, i.e., CCD841 and HEM there was a ~22% enhancement of the gap enclosure rate. The micrographs of scratch test for primary CCD841 and HEM cells compared with their cancerous counterparts SW48 and MM418-C1 are shown as Appendix A.

### 2.8. Effect of AuNPs and IR on Cell Adhesion

The combined effect of IR and/or AuNPs on SW48, CCD841, MM418-C1 and HEM cell adhesiveness was examined by treating the cells with 1 mM AuNPs 24 h prior to being exposed to two different doses (2 and 5 Gy) of 6 MV X-rays. Adherent cells grown in tissue culture were trypsinised and the cells plated out into a 6-well plate. After 4 h incubation, the wells were gently washed with phosphate-buffered saline (PBS) and the number of attached cells in a defined area (0.25 × 0.25 mm or 62,500 µm^2^) was counted.

Treatment of the colon epithelial-derived cells (SW48 and CCD841 cells) with AuNPS had no effect on cell adhesion when compared to untreated controls (Figure 8A,B). When the SW48 colorectal adenocarcinoma cells were exposed to 2 or 5 Gy X-rays there was a dose-dependent increase in the number of attached cells (217% and 280%, respectively) compared to the untreated controls (Figure 8A). However, when the SW48 cells were pretreated with AuNPs and exposed to either 2 or 5 Gy X-rays no difference in cell adherence was seen compared to those cells only exposed to IR. Treatment of the CCD841 colon epithelial cells with either AuNPs, IR or AuNPs + IR had no effect on the attachment of these cells when compared to the untreated controls (Figure 8B).

Treatment of the melanocytic cells (MM418-C1 and HEM cells) with AuNPS had no effect on cell adhesion when compared to untreated controls (Figure 9A,B). This result was similar to that seen in the colonic-derived cells. When the MM418-C1 cells were exposed to 2 or 5 Gy X-rays there was a dose-dependent increase in the number of attached cells (205% and 261%, respectively) compared to the untreated controls (Figure 9A). However, when the MM418-C1 cells were pretreated with AuNPs and exposed to either 2 or 5 Gy X-rays no significant difference in cell adherence was seen compared to those cells only exposed to IR. Treatment of the HEM cells with either AuNPs, IR or AuNPs + IR had no effect on the attachment of these cells when compared to the untreated controls (Figure 9B).

## 3. Discussion and Conclusions

The majority of cancers (~90%) originate in epithelial tissues [16] and most of the cancer-related deaths are caused by metastasis [17]; therefore, understanding the role played by cell migration in epithelial cells is essential to enhance the effectiveness of IR in treating such tumours. The epithelium functions as a protective layer which is actively involved in the wound healing process [18]. It is shown that normal cell behavior during the wound healing (migration) is similar to cancer metastasis and can be considered as two sides of the same coin [18,19,20]. The similarity between wound healing and cancer metastasis is “vital” for radiation therapy because those non-cancerous cells surrounding the treatment area (along with cancer cells) are exposed to the same therapeutic radiation dose. In this in vitro study, it was anticipated that normal epithelial cell lines, i.e., CCD841 and HEM and their respective cancerous counterparts, i.e., SW48 and MM418-C1 cells after exposure to a clinical dose (5 Gy of X-ray) should respond differently with respect to cell migration and adherence. In our previous study we observed that pretreating the cells with 1 mM AuNPs 24 h prior to exposure to IR, while having minimal effects on cell viability did inhibit cell migration [14], in this study, we examined the combined effects of AuNPs and IR on the viability and motility of two related cell lines that originate from the colon (CCD841 and SW48 cells) and the epidermis (HEM and MM418-C1 cells), to observe if these effects are tumour specific.

All cell lines incorporated gold intracellularly following exposure to AuNPs over a 24 h period (Figure 1). While the levels taken up by the different cell lines were similar at concentrations < 1 mM, however, at higher concentrations (>1 mM) the tumour cell lines (SW48 and MM418-C1) did not incorporate any further gold in their cells, unlike that seen in their corresponding primary cell lines (CCD841 and HEM). Ivošev et al. [21] also observed an increased uptake of gold from AuNPs in a range of tumour cell lines when compared to dermal fibroblasts, however they did not compare matched cell types (cancerous vs non-cancerous cells from the same cell type) as we did in this study. They observed that these tumours cells had different internalization pathways which may have explained the observed differences. Further studies using other matched cell lines from different regions of the body are warranted to observe if tumour cells take up more or less NPs that their corresponding non-cancerous cohorts. The results of which could have an impact on patients who are given AuNPs prior to exposure to therapeutic IR doses.

When the cells were treated with AuNPs, we observed no decrease in cell viability following incubated with 0–4 mM AuNPs over 48 h (Figure 2). It was seen from Figure 1, that at concentrations greater > 1 mM, the level of gold within the non-cancerous cells was proportional to the external concentration, however in the tumour cells intracellular levels did not increase above this concentration. These results are in agreement with that reported by Trono et al. [22]. They observed that uptake and incorporation of AuNPs depend on various factors such as cell type (non-cancerous or cancerous), the concentration and size of the nanoparticles and the incubation period with the nanoparticles. In order to ensure that the internal concentration of gold was not having a direct effect on our studies all cell lines were treated with 1 mM AuNPs. When cells were pretreated with 1 mM AuNPs for 24 h and then subsequently exposed to 0–5 Gy X-rays (Figure 4) there was no decrease in viability after 48 h when compared to those cells who were only exposed to IR (Figure 3). A similar result was also observed when human prostate DU145 and lung A549 cancer cells were exposed to IR in the presence of absence of 1 mM AuNPs [14].

Exposure to increasing doses of IR (0–6 Gy) resulted in a reduction in the viability of all four cell types over 48 h (Figure 3). Similar effects were observed when DU145 prostate and A549 lung cancer cells were exposed to IR [14]. The cytotoxic effects of IR were more pronounced on the tumour cells (SW48 and MM418-C1 cells) when compared to their non-cancerous counterparts (CCD841 and HEM). The difference in cell viability between these pairs of cells was ~15% following exposure to high doses of X-rays (e.g., the viability of HEM and MM418-C1 cells following exposure to 6 Gy was 78% and 60%, respectively). A similar result was also seen in CCD841 and SW48 exposed to the same IR dose resulting in 79% and 65% cell viability, respectively.

Different response to the IR and/or AuNPs was observed in cancer cells compared to that of their normal counterparts with regards to their migration rates. The migration of the tumour cell lines (SW48 and MM418-C1) were retarded when these cells were treated with either IR and/or AuNPs, while that of their non-cancerous counterparts (CCD841 and HEM) were enhanced when given the same treatments (Figure 5, Figure 6 and Figure 7). This finding is in line with Moncharmont et al. [5] who reported that IR can enhance or diminish cell migration in a range of tumour cell lines. The effects of AuNPs and that of IR on retarding the migration of both tumour cells (SW48 and MM418-C1) were similar (15–20%), however an additive effect (~35%) was seen when these cells were exposed to both treatments. This finding was similar to that seen when DU145 prostate and A549 lung cancer cells were treated with AuNPs and or IR [14]. Of interest was that when the corresponding non-cancerous cells (CCD841 and HEM) were exposed to the same treatments cell migration was enhanced. In these cells, both AuNPs and IR individually enhanced cell migration by 13–16% but when they were exposed to both treatment migration increased by ~22%.

While both AuNPs and IR treatments impaired the migration of the tumour cells compared to their non-cancerous counterparts, AuNPs had no effect on adhesiveness of these cell lines (Figure 8 and Figure 9). IR enhanced the adhesiveness of the tumour cells (SW48 and MM418-C1 both ~230%) compared to their non-cancerous counterparts and this effect was proportional to the X-ray dose. This was similar to that seen when H1299 human non-small cell lung cancer cells [23] and DU145 human prostate and A549 lung cancer cells [14] were exposed to IR. When these tumour cells were pretreated with AuNPs no additive effect was observed in either the SW48 and MM418-C1 cells, which was similar to that seen in DU145 human prostate and A549 lung cancer cells [14]. These changes to cell adhesion appear to be primarily mediated by IR as AuNPs had no effect on this process. A summary of the effects of AuNPs and IR on the migration and adhesiveness of the cells used in this study is shown in Table 1.

The differences observed between the tumour cells and their non-cancerous counterparts do not appear to be related to acquired mutations. SW48 cells are colorectal adenocarcinoma cells and like the CCD841 colon epithelial cells they do not carry mutations for KRAS, NRAS, BRAF and PI3KCA [24]. MM418-C1 cells possess the BRAF^V600E^ mutation and are homozygous for *CDKN2A* deletions [25], while HEM primary melanocytes do not carry these mutations.

Focal adhesions, podosomes and invadosomes located on the cell membrane play a major role in cell migration and adherence [26]. These structures integrate external signals which cause cells to alter their morphology, so they can migrate and adhere to the substrate. Tsutsumi et al. [23] observed that H1299 irradiated cells were more adhesive than unirradiated cells, which was due to increased numbers of focal adhesions on the cell membrane. They also observed an increase in matrix metalloprotease activity in the irradiated cells which would enhance cell migration. Following exposure to IR increased levels of phosphorylated focal adhesion kinases (FAK) as well as p38 and JNK was observed in lung A549 cancer cells [27]. The addition of FAK inhibitors reduced the migration of radiation-induced medulloblastoma cells confirming the role these kinases play in IR-induced migration in these tumour cells, but not in non-cancerous cells. Further studies on the role played by FAK, p38 and JNK are warranted to see if there is a different response elicited between cancerous and non-cancerous cells when exposed to IR.

AuNPs on the other hand were shown to have a similar effect to that of IR on cell migration. Of interest that while it reduced tumour cell migration it enhanced that of the corresponding non-cancerous cells. Previous studies have shown the inhibitory effects of AuNPs on the migration of HEY A8 ovarian cancer cells [6], DU145 human prostate and A549 lung cancer cells [14]. As AuNPs are taken up into the cell, it has been shown that in HEY A8 cells they were trapped in the nuclear membrane which increased its stiffness which in turn reduced cell migration [6], however it is unknown whether these particles have the same effect on non-cancerous cells. Recently, in examining mouse macrophages and mesenchymal stem cells, AuNPs interfered with the podosomes of these cells resulting reduced focal adhesions and ECM degradation in the macrophages while the opposite effect was seen in the stem cells [28]. This cell specific effect of AuNPs on cell migration needs to be further investigated as we believe this is the first study to compare cancerous and non-cancerous cells of the same lineage, and highlights that differences exist between them. While we observed no effects of AuNPs on cell adhesion, Lo et al. [29] found that naked AuNPs could reduce the adhesiveness of vascular smooth muscle cells by inhibiting FAK phosphorylation and actin cytoskeletal reorganisation. The authors used a range of AuNPs prepared by chemical or physical methods and this effect may be related to the nanoparticle itself. Further studies are warranted to investigate the effects AuNPs have on focal adhesions and podosomes as well as cell signalling pathways to determine how they interact with these cells. Similarly undertaking proteomic studies to observe what changes have occurred in the cell as a result of AuNPs and/or IR treatment is warranted, and along with the studies on changes in the activities of membrane signalling pathway intermediates will form the basis of suture studies. The main aim of this study was to investigate the combined effects of IR and AuNPs on two types of epithelial cells (colon—CCD841 and skin—HEM) and their cancerous counterparts (SW48 and MM418-C1).

Our results decisively showed that the therapeutic dose of IR separately and in combination with AuNPs significantly reduced the viability and migration rate of the tumour cells while having minor negative effects on the proliferation of their corresponding non-cancerous cohort while at the same time enhancing their migration rate. These findings are consistent with the effectiveness of radiation therapy in overall tumour control and highlight the benefits of utilizing AuNPs as a nontoxic and injectable agent in radiation therapy to improve its therapeutic outcomes. One of the major challenges facing radiotherapy is cancer metastasis, which are caused by cell migration. Hence, understanding testing, monitoring, and controlling metastasis could potentially result in more efficient radiotherapy. This research presents an efficient way of reducing cancer cell’s migration which could be utilised for controlling metastasis. Therefore, beside what has been well established about the role of NPs in enhancing radiation effects “dose” this work shows another role for such particles in radiotherapy and that is in reducing cancer cells migration which could lead to less possibilities for cancer metastasis. This could lead to vital changes in the regimen of radiation dose delivery in radiotherapy.

## 4. Materials and Methods

### 4.1. Cell Culture

Human colorectal adenocarcinoma cells (SW48: ATCC^®^ CCL-231^TM^) and primary human colon epithelial cells (CCD841: ATCC^®^ CRL-1790^TM^) were purchased from ATCC (Manassas, VA, USA). Human primary melanoma (MM418-C1) cells were a gift from Dr Glen Boyle (QIMR, Brisbane, Australia) while primary human epithelial melanocytes (HEM: ATCC^®^ PCS-200-013^TM^) were purchased from ATCC and were used in this study.

SW48 cells were cultured and maintained in Leibovitz’s L-15 (Gibco^®^, Grand Island, NY, USA), 10% foetal bovine serum (FBS) (Gibco^®^) and 1% Penicillin-Streptomycin (Gibco^®^). CCD841 cells were cultured and maintained in EMEM (Gibco^®^), 10% FBS and 1% Penicillin-Streptomycin. MM418-C1 and HEM cells were cultured and maintained in RPMI 1640 (Gibco^®^), 10% FBS and 1% Penicillin-Streptomycin.

The CCD841, MM418-C1 and HEM cells were incubated at 37 °C with 5% CO_2_ in a humidified environment, while SW48 cells were maintained at 37 °C in a humidified environment and in a gas free exchange with atmosphere as CO_2_ and air mixture is detrimental to this cell line.

### 4.2. AuNPs Preparation

Spherical AuNPs were purchased from Nanoprobes (Yaphank, NY, USA). These AuNPs possess a metal core diameter of ~15 nm, which has been stabilized with a highly water-soluble thiol-based ligand [30]. The AuNPs were stable in the culture media used in these experiments and no aggregation of the nanoparticles was observed after 48 h when media samples were observed under light microscopy using a 40× objective lens. The AuNPs solution was diluted using the appropriate cell culture medium to a final concentration of 0.197 mg/mL.

### 4.3. Inductively Coupled Plasma Mass Spectrometry (ICP MS) Measurement of Celluar AuNPs

Intracellular Au levels were determined using a ICP MS (Agilent 7700, Santa Clara, CA, USA). The cells (SW48, CCD841, MM418-C1 and HEM) were seeded (10^6^ cells/well), in 6-well plates and incubated at 37 °C with 5% CO_2_ in a humidified environment for 24 h. After which, the cells were exposed to 0 to 4.0 mM AuNPs for 24 h. At the end of this period the cells were gently washed with warm PBS (37 °C), and then trypsinised. An aliquot of the resuspended cells was counted using a haemocytometer under a light microscope. The rest of the resuspended cells were dissolved in 1% HCl. In the ICP MS a calibration curve from a known gold standard (TraceCERT^®^, Sigma-Aldrich, St Louis, MO, USA) was established, and from which, the cellular uptake of the AuNPs was calculated and expressed as pg Au/cell.

### 4.4. Viability Assay

Cells (3 × 10^3^ cells/well) were seeded in a 96-well plate incubated at 37 °C with 5% CO_2_ in a humidified environment. After 24 h, the cells were treated with various concentrations of AuNPs ranging from 0 to 4.0 mM and/or exposed to 0 to 6 Gy of 6 MV X-rays.Cell viability was determined 48 h after treatment with AuNPs and/or IR, using the MTS assay as described previously [14].

### 4.5. Cell Irradiation

The cells were irradiated with 6 MV X-ray generated by Linac (Elektra Synergy, Stockholm, Sweden) located at Australian Radiation Protection and Nuclear Safety Agency (ARPANSA), Yallambie, Australia. The radiation was delivered as a single fraction for each dose, i.e., 0 Gy for control groups and 1 to 6 Gy for treated groups, as described previously [14]. The delivered dose was calculated from measurements using an ion chamber before the cell irradiation.

### 4.6. Scratch Assay

Cells were cultured in 6-well plates for 24 h until they reached ~90% confluency. The cells were washed with warm (37 °C) PBS before a sterile (200 μL) yellow pipette tip was used to create a ~600 µm gap. After this the wells were washes with warm PBS, and and fresh tissue culture media added. Gap filling was measured using the CytoSmart^®^ Live Image System (Piet Heinstraat, Zutphen, Holland) [12]. The gap area at 0 h and 24 h were measured with ImageJ^®^ software and the relative migration was calculated using Equation (1) as seen below.
(1)$$Relative\;Migration=\frac{Gap\;area\;at\;0\;h-Gap\;area\;at\;24\;h\;\left(Untrated\;control\;cells\right)}{Gap\;area\;at\;0\;h-Gap\;area\;at\;24\;h\;\left(Treated\;cells\right)}$$

### 4.7. Adhesion Assay

Cellular adhesion (attachment) to the polystyrene surface of the well of a 6-well plate was evaluated in both control and treated groups. Cells grown in in 25 cm^2^ flasks were treated with either 1 mM AuNPs or exposed to 2 and/or 5 Gy 6 MV X-ray and incubated for 24 h. Where the cells were exposed to both treatments they were initially exposed to 1 mM AuNPs for 24 h before being irradiated with either 2 and/or 5 Gy 6 MV X-ray and incubated for a further 24 h. At the end of this period, each experimental group, i.e., control (untreated), irradiated with 2 or 5 Gy and/or treated with AuNPs were trypsinised and 2 × 10^4^ cells seeded in 6-well plates. After 4 h incubation at 37 °C, the non-adherent cells were gently washed with warm PBS (37 °C) and images were taken using an EVOS^®^ XL Cell Imaging System (Thermo Fisher Scientific, Waltham, WA, USA). The adherent cells in four different random 6.25 × 10^4^ µm^2^ areas (250 × 250 µm square region) were determined and the average values plotted.

### 4.8. Statistical Analysis

All presented data within this paper are the mean of at least three independent experiments. Statistical comparison between the control group and AuNPs group, IR group, and AuNPs + IR group were performed using one-way analysis of variance (ANOVA) with IBM SPSS Statistics version 25 (IBM Australia Ltd., St Leonards, NSW, Australia). Results are reported as mean ± standard error of the mean (SEM). * *p* < 0.05 were considered statistically significant.

## Figures and Tables

**Figure 1 ijms-22-01418-f001:**
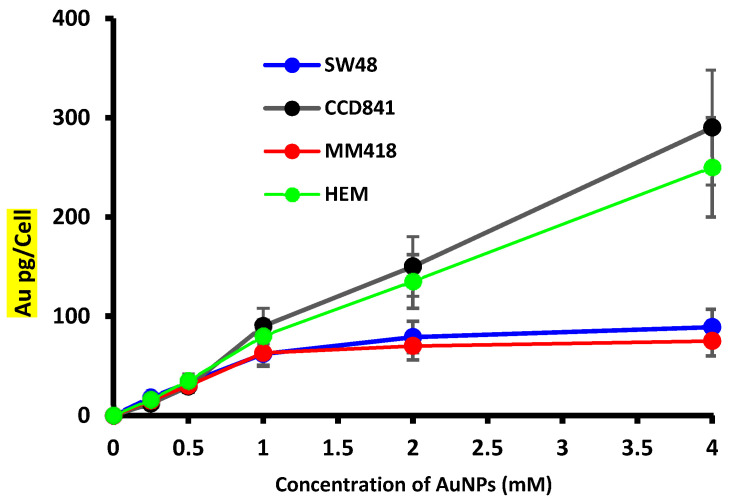
Effect of external AuNPs concentration on the accumulation of gold in human colon adenocarcinoma (SW48) and melanoma (MM418-C1) cell lines compared to their non-cancerous counterparts (CCD841 and HEM cells). The cells were exposed to 0–4 mM AuNPs for 24 h, and the level of Au in the cells were determined using inductively coupled plasma mass spectrometry (ICP MS). The ICP MS intensities were converted to mass of AuNPs per cell by using Au ion standard curve and cell counts. Results are expressed as the mean ± SEM of 3 replicates.

**Figure 2 ijms-22-01418-f002:**
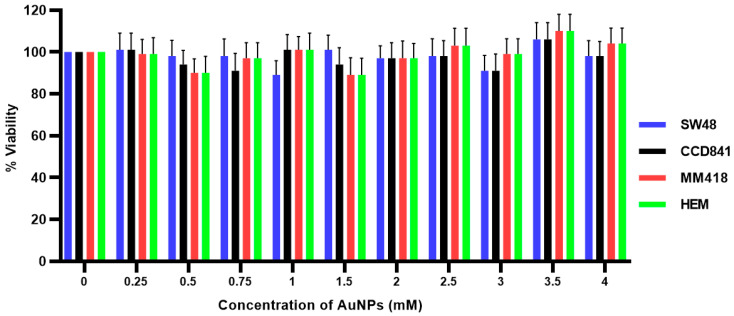
Effect of AuNPs on the viability of human colon adenocarcinoma (SW48) and melanoma (MM418-C1) cell lines compared to their non-cancerous counterparts (CCD841 and HEM cells). The cells were treated with 0–4 mM AuNPs for 48 h and cell viability (viability%) was determined using the MTS assay. No significant differences in cell viability were seen in all cell types at different concentration of AuNPs (*p* > 0.05). Results expressed are the mean ± SEM of 3 replicates.

**Figure 3 ijms-22-01418-f003:**
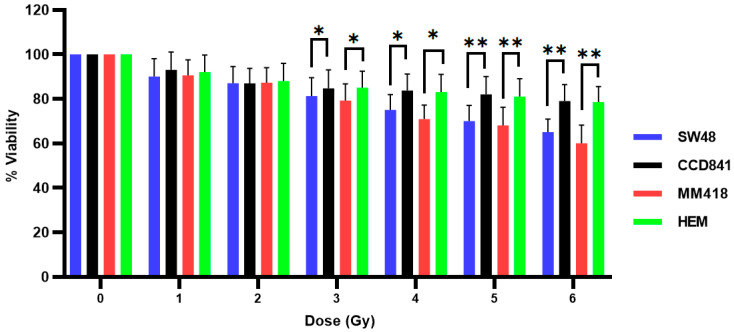
Effect of ionizing radiation (IR) on the viability of human colon adenocarcinoma (SW48) and melanoma (MM418-C1) cell lines compared to their non-cancerous counterparts (CCD841 and HEM cells). The cells were irradiated with 0–6 Gy X-rays and viability % was measured 48 h post-irradiation using the MTS assay. Results are expressed as the mean ± SEM of 3 replicates. Significance of different treatments compared to control is represented by black lines shown as * *p* < 0.05 and ** *p* < 0.01.

**Figure 4 ijms-22-01418-f004:**
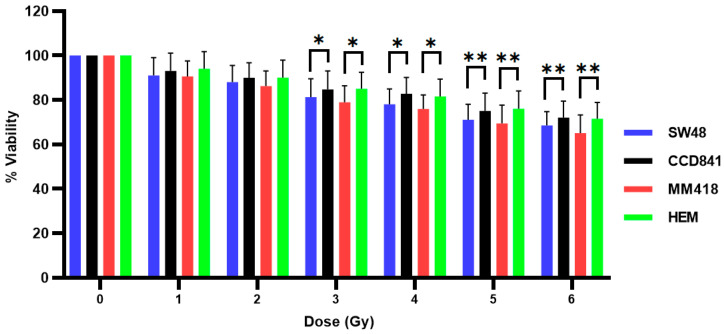
Effect of AuNPs + IR on the viability of human colon adenocarcinoma (SW48) and melanoma (MM418-C1) cell lines compared to their non-cancerous counterparts (CCD841 and HEM cells). The cells were treated with 1 mM AuNPs for 24 h prior to being exposed to 0–6 Gy of 6 MV X-rays. Cell viability (% viability) was measured 48 h post-irradiation using the MTS assay. Results are expressed as the mean ± SEM of 3 replicates. Significance of different treatments compared to control is represented by black lines shown as * *p* < 0.05 and ** *p* < 0.01.

**Figure 5 ijms-22-01418-f005:**
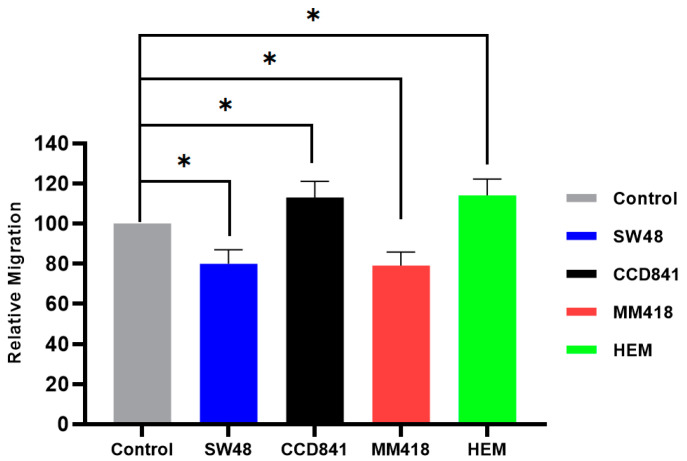
The effect of AuNPs on the migration of human colon adenocarcinoma (SW48) and melanoma (MM418-C1) cell lines compared to their non-cancerous counterparts (CCD841 and HEM cells). The cells were treated with 1 mM AuNPs 24 h prior to the scratch test. Control represents the corresponding untreated cells in which the size of the gap closure over 24 h was expressed as 100%. Results were calculated as a percentage of that of the corresponding untreated controls over the 24 h period and are expressed as the mean ± SEM of 3 replicates. Significance of different treatments compared to the untreated control is represented by black lines shown as * *p* < 0.05.

**Figure 6 ijms-22-01418-f006:**
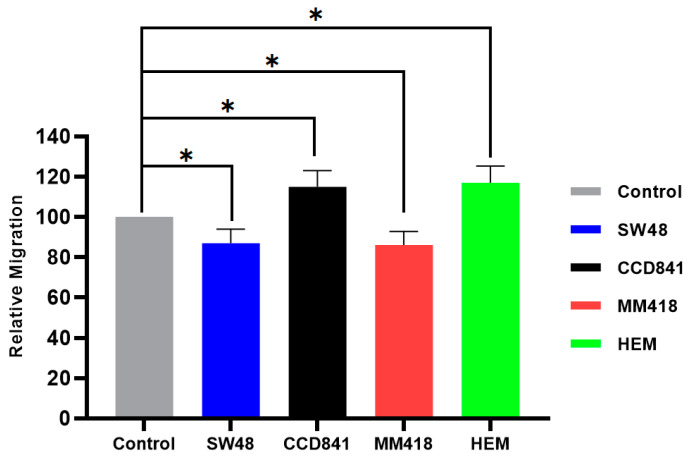
The effect of IR on the migration of human colon adenocarcinoma (SW48) and melanoma (MM418-C1) cell lines compared to their non-cancerous counterparts (CCD841 and HEM cells). The cells were irradiated with 5 Gy X-ray 24 h prior to the scratch test. Control represents the corresponding untreated cells in which the size of the gap closure over 24 h was expressed as 100%. Results were calculated as percentage of the corresponding untreated controls over that 24 h period and are expressed as the mean ± SEM of 3 replicates. Significance of different treatments compared to the untreated control is represented by black lines shown as * *p* < 0.05.

**Figure 7 ijms-22-01418-f007:**
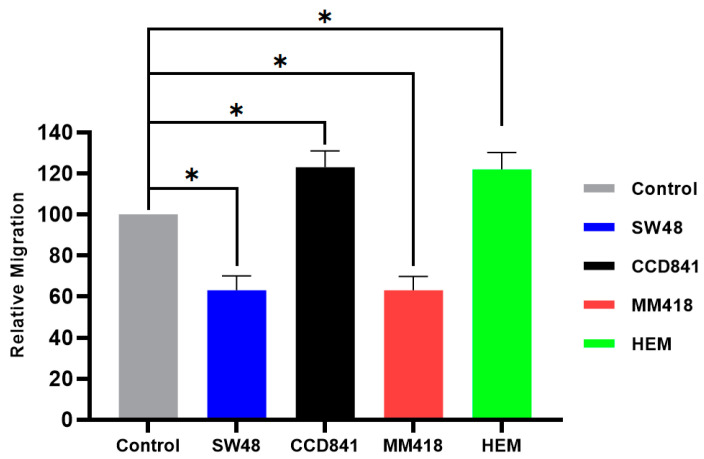
The effect of AuNPs + IR on the migration of human colon adenocarcinoma (SW48) and melanoma (MM418-C1) cell lines compared to their non-cancerous counterparts (CCD841 and HEM cells). The cells were irradiated with 5 Gy X-ray 24 h prior to the scratch test. Control represents the corresponding untreated cells in which the size of the gap closure over 24 h was expressed as 100%. Results were calculated as percentage of the corresponding untreated controls over that 24 h period and are expressed as the mean ± SEM of 3 replicates. Significance of different treatments compared to the untreated control is represented by black lines shown as * *p* < 0.05.

**Figure 8 ijms-22-01418-f008:**
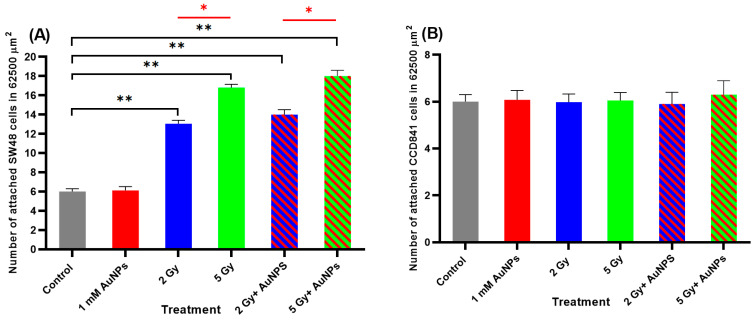
Effect of AuNPs and/or IR on the adhesion of (**A**) SW48 and (**B**) CCD841 cells in Vitro. Cells were treated with 1 mM AuNPs 24 h and/or 2 and 5 Gy (6 MV X-rays) and after 24 h the cells were trypsinised and plated in 6-well plates. After 4 h, the number of adhered cells in a 62,500 µm^2^ area were counted. Results are expressed as the mean ± SEM of 3 replicates. Significance of different treatments compared to the untreated control is represented by black lines shown as ** *p* < 0.01 and red lines shown as * *p* < 0.05.

**Figure 9 ijms-22-01418-f009:**
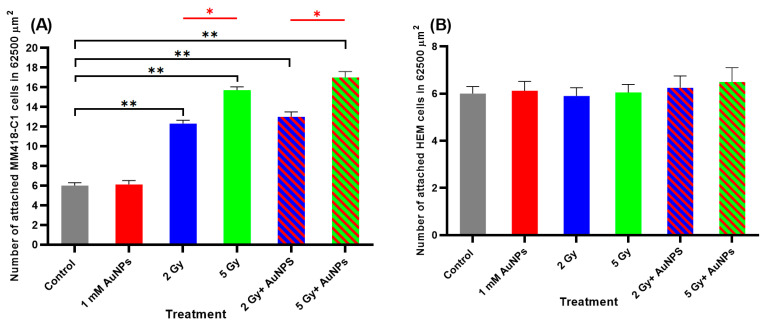
Effect of AuNPs and/or IR on the adhesion of (**A**) MM418-C1 and (**B**) HEM cells in Vitro. Cells were treated with 1 mM AuNPs 24 h and/or 2 and 5 Gy (6 MV X-rays) and after 24 h the cells were trypsinised and plated in 6-well plates. After 4 h, the number of adhered cells in a 62,500 µm^2^ area were counted. Results are expressed as the mean ± SEM of 3 replicates. Significance of different treatments compared to the untreated control is represented by black lines shown as ** *p* < 0.01 and red lines shown as * *p* < 0.05.

**Table 1 ijms-22-01418-t001:** The effect of IR and/or AuNPs on the migration and adhesion of SW48, CCD841, MM418-C1 and HEM cells. Changes due to treatment are shown as an increased effect (↑), decreased effect (↓) or no effect (–).

	Effect of Treatment on Cell Migration			Effect of Treatment on Cell Adhesion		
Cell line	AuNPs	IR	IR+AuNPs	AuNPs	IR	IR+AuNPs
**SW48**	↓	↓	↓	–	↑	↑
**CCD841**	↑	↑	↑	–	–	–
**MM418-C1**	↓	↓	↓	–	↑	↑
**HEM**	↑	↑	↑	–	–	–

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
