# Peer review of "Differential Effects of Gold Nanoparticles and Ionizing Radiation on Cell Motility between Primary Human Colonic and Melanocytic Cells and Their Cancerous Counterparts"

_ijms, 2021, doi:10.3390/ijms22031418_

Round 1
Reviewer 1 Report
The authors present a study on the adhesion and migration of two normal and two cancer cell lines upon X-ray irradiation and AuNP incubation. The work is well structured, and the conclusions are supported by the results. This work is important to the scientific community.
Regarding the novelty, the authors have recently published a similar paper, which they cite, where they present the same study on two other cancer cell lines. I believe most of the conclusions on this work lose their impact since they were previously reported on other cell lines. However, I find it very interesting that in this paper the authors compare the cancer cell lines with their healthy counterparts.
Suggestions of improvement:
- Include a figure showing the scratch assay photos.
- In section 4.3 could be more detailed, it is not clear if a PBS wash was performed before HCL addition. This would prevent detecting Au from the media.
- Collect proteomics data before and after irradiation and incubation with AuNP to understand the mechanism behind the migration effect observed.
With no other scientific remark on this work, I recommend it for publication.
Author Response
Dear Reviewer-1,
Thanks for the positive views on our manuscript. We highly appreciate your time. Regarding your valuable comments here are our replies:
- Include a figure showing the scratch assay photos.
Answer: The requested photos/figures have been added to the manuscript as supplementary Figure S1 and S2.
- In section 4.3 could be more detailed, it is not clear if a PBS wash was performed before HCL addition. This would prevent detecting Au from the media.
Answer: More details have been added to section 4.3 (Line 477-478).
- Collect proteomics data before and after irradiation and incubation with AuNP to understand the mechanism behind the migration effect observed.
Answer: It is a valuable comment; however, to perform these experiments and include the results in this manuscript it will take a long time and will make the manuscript too long. We plan to do this step in our next work so thanks for this valuable suggestion.
Reviewer 2 Report
Report of Manuscript ijms-1092985 for IJMS
Title: Differential effects of Gold Nanoparticles and Ionizing Radiation on cell motility between primary human colonic and melanocytic cells and their cancerous counterparts by Elham Shahhoseini et al.
This referred manuscript aims to determine the effects of gold nanoparticles and ionizing radiation, individually and in combination on the viability and motility of human primary cells and their cancerous counterparts. The authors have used different cell types.
The paper is well-written, the English is good and I think that this work could be of interest for the field of basic and applicative research of nanoparticles in radiation therapy. The paper is technically sound, but it is specific interest.
The work is well structured and the proposed goals were achieved. The manuscript contains new information to justify publication. The methods described comprehensively. The list of references should be improved and modified.
The interpretations and conclusions justified by the results. Different and complementary analyzes were carried out. The manuscript appears complete in all its parts.
The paper has a good general aspect, but some arguments and technical details are required.
Main Text - Different minor typo-corrections that should be performed.
Title - I suggest avoiding capital letters when not needed.
Abstract - I think this abstract is excessively long (about 400 words). In general, an abstract should be concise and incisive. The authors must shorten it following the journal's standards.
Keywords – I suggest to ad “radiotherapy”
Introduction - I suggest that the authors expand the introduction. There are also combined therapies between nanoparticles and radiotherapy, nanoparticles and magnetic hyperthermia, nanoparticles and phototherapy for the treatment of cancer. This research field is very vast and of great scientific interest. The authors must increase references. I suggest for example to add “https://doi.org/10.3390/nano10101919”; “https://doi.org/10.3390/app10207322” https://doi.org/10.3390/nano10112310 and many others.
- Fig.1 - The title of the y-axis overlaps the scale. Change please.
- It is unclear whether the average diameter in the nanoparticles was measured.
- Line 489 – The authors write “no aggregation were observed within 48 h”.
- How was this assessment carried out?
- Did the authors use techniques such as DLS?
Please justify.
- Line 505 - lead to the top row.
- Section 4.5 - How did the authors assess the dose imparted? It is unclear what is the measurements uncertainty. Do the authors use TPS as reference dosimeter or ionization chambers?
- The conclusions need to be enriched to emphasize the applicability of the results found: this aspect is fundamental to the publication and impact of this manuscript.
Other than these minor comments, I feel that this is a well-designed and good study. This reviewer hopes to receive a new and improved version of the manuscript.
Author Response
Dear Reviewer-2,
Thanks for the positive views on our manuscript. We highly appreciate your time. Regarding your comments here are our replies:
- The list of references should be improved and modified.
Answer 1: Thanks for the observation, the list of references has been improved & modified.
- Main Text - Different minor typo-corrections that should be performed.
Answer 2: Thank you for your observation and this has been accomplished throughout the whole manuscript.
- Title - I suggest avoiding capital letters when not needed.
Answer 3: Thanks for the comment, the tile has been rewritten.
- Abstract - I think this abstract is excessively long (about 400 words). In general, an abstract should be concise and incisive. The authors must shorten it following the journal's standards.
Answer 4: We agree, the abstract has been shortened to about 200 words to meet the journal standards & requirements.
- Keywords – I suggest to ad “radiotherapy”
Answer 5: Thank you for the suggestion, the word ‘Radiotherapy’ has been added to the keyword section.
- Introduction - I suggest that the authors expand the introduction. There are also combined therapies between nanoparticles and radiotherapy, nanoparticles and magnetic hyperthermia, nanoparticles and phototherapy for the treatment of cancer. This research field is very vast and of great scientific interest. The authors must increase references. I suggest for example to add “https://doi.org/10.3390/nano10101919”; “https://doi.org/10.3390/app10207322” https://doi.org/10.3390/nano10112310 and many others.?
Answer 7: Thanks for the suggestion, a sentence has been added to the Introduction section (Line 49- 51) and the suggested references have been added as requested.
- 1 - The title of the y-axis overlaps the scale. Change please.
Answer 7: Thank for the correction, the y-axis in Figure 1 has been modified.
- It is unclear whether the average diameter in the nanoparticles was measured.
Answer 8: Thanks for this comment. Since the diameter of the same commercial gold nanoparticles has been measured by our research lab [1] and others [2] using TEM previously and as these results were found to be consistent with the supplier’s provided information, measuring the diameter of the NPs has/was not been performed for this work.
[1] Smith, C. The investigation of alanine/EPR dosimetry, dose enhancement caused by AuNPs and the novel synthesis of bimetallic-nanoparticles via neutron capture. Dissertation, RMIT University, Melbourne, 2017.
[2] Bahamonde, J.; Brenseke, B.; Chan, M.Y.; Kent, R.D.; Vikesland, P.J.; Prater, M.R. Gold Nanoparticle Toxicity in Mice and Rats: Species Differences. Toxicologic pathology 2018, 46, 431-443, doi:10.1177/0192623318770608.
- Line 489 – The authors write “no aggregation were observed within 48 h”. How was this assessment carried out? Did the authors use techniques such as DLS (dynamic light scattering)? Please justify.
Answer 9: Thanks for the comment, we did not use DLS in our work. However, the aggregation was assessed by microscopic visual inspection. This has been added into the manuscript Section 4.2. (Line 471).
- Line 505 - lead to the top row.
Answer 10: Done.
- Section 4.5 - How did the authors assess the dose imparted? It is unclear what is the measurements uncertainty. Do the authors use TPS as reference dosimeter or ionization chambers?
Answer 11: ARPANSA as Primary Standard Dosimetry Laboratory for Australia and New Zealand, has done the irradiation for this work. This lab uses ionisation chambers for radiation dosimetry with the uncertainty of the dose delivery being <5% (Line 493-494).
- The conclusions need to be enriched to emphasize the applicability of the results found: this aspect is fundamental to the publication and impact of this manuscript.
Answer 12: The conclusion has been revised and applicability of the results has been emphasized. (Line 442-450)
Round 2
Reviewer 2 Report
All my previous comments have been carefully addressed by the authors, and the manuscript has been properly modified accordingly.
The reviewer doesn't have additional comments.
Thanks to the authors for working on this revision.